# High-fidelity DNA replication in *Mycobacterium tuberculosis* relies on a trinuclear zinc center

Soledad Baños-Mateos[1], Anne-Marie M. van Roon[1], Ulla F. Lang[1], Sarah L. Maslen[1], J. Mark Skehel[1] & Meindert H. Lamers [1]

High-fidelity DNA replication depends on a proofreading 3′–5′ exonuclease that is associated with the replicative DNA polymerase. The replicative DNA polymerase DnaE1 from the major pathogen *Mycobacterium tuberculosis* (Mtb) uses its intrinsic PHP-exonuclease that is distinct from the canonical DEDD exonucleases found in the *Escherichia coli* and eukaryotic replisomes. The mechanism of the PHP-exonuclease is not known. Here, we present the crystal structure of the Mtb DnaE1 polymerase. The PHP-exonuclease has a trinuclear zinc center, coordinated by nine conserved residues. Cryo-EM analysis reveals the entry path of the primer strand in the PHP-exonuclease active site. Furthermore, the PHP-exonuclease shows a striking similarity to *E. coli* endonuclease IV, which provides clues regarding the mechanism of action. Altogether, this work provides important insights into the PHP-exonuclease and reveals unique properties that make it an attractive target for novel anti-mycobacterial drugs.

[1] MRC Laboratory of Molecular Biology, Francis Crick Avenue, Cambridge Biomedical Campus, Cambridge CB2 0QH, UK. Correspondence and requests for materials should be addressed to M.H.L. (email: mlamers@mrc-lmb.cam.ac.uk)

Faithful replication of genomic DNA is essential to all forms of life. Three universally conserved mechanisms prevent the generation of mutations during the replication process (reviewed in Kunkel[1]). High-fidelity DNA polymerases accurately replicate the DNA, with error rates as low as $10^{-6}$ per base pair (bp). In addition, all replicative DNA polymerases contain, or bind to, a 3′–5′ exonuclease that removes any misincorporated nucleotides, further reducing the error rate of DNA replication by a factor of 10–100. Finally, the DNA mismatch repair system detects and removes any remaining replication errors, lowering the overall error rate of DNA replication to $10^{-10}$. Surprisingly, *Mycobacterium tuberculosis* (Mtb), and indeed all mycobacteria lack DNA mismatch repair[2], yet do not show elevated levels of mutagenesis when compared to *Escherichia coli* (*E. coli*)[3]. This suggests that the replication machinery itself may be capable of unusually high-fidelity DNA replication, but the mechanisms by which this is achieved are not known. Understanding the mechanisms that control mutation rates in mycobacteria is crucial as point mutations drive antibiotic resistance in tuberculosis (TB)[3, 4], which poses a major threat to global health[5].

Recently, we have shown that Mtb does not use the canonical DnaQ-like DEDD exonuclease that is used in *E. coli* DNA polymerases I, II, and III, as well as the eukaryotic replicative DNA polymerase pol δ and pol ε. Instead, the Mtb replicative DNA polymerase DnaE1 uses its polymerase and histidinol phosphatase (PHP) domain as replicative exonuclease[6]. This domain is present in all bacterial replicative DNA polymerases, but not in the eukaryotic replicases[7]. The function of the PHP domain, however has long been ill defined. It has been suggested that it may function as a pyrophosphatase that removes the by-product of DNA replication[7, 8], whereas for *E. coli* it has been shown to stabilize the polymerase domain[9]. Interestingly, in some bacteria it has been shown that the PHP domain retains 3′–5′ exonuclease activity, although the role of this activity in DNA replication was unclear[10, 11]. Recently, we have shown that in mycobacteria, the PHP domain is the essential replicative exonuclease and that the DnaQ-like DEDD exonuclease in contrast is dispensable[6]. Sequence analysis furthermore revealed that the PHP-exonuclease is the most abundant replicative exonuclease in bacterial replicases, and that it may be the ancestral proofreader[6]. In addition, we have shown that the PHP domain of Mtb DnaE1 is a potential target for new antibiotics as its inactivation leads to strongly decreased growth rates and renders mycobacteria sensitive to nucleotide analogs that are used in antiviral therapy.

To gain insight into the mechanism of action of the DnaE1 PHP-exonuclease, we have determined the crystal structure of Mtb DnaE1 to 2.8 Å resolution. The PHP domain shows a nine-residue active site that coordinates three zinc ions. Furthermore, cryo-electron microscopy (cryo-EM) analysis of DnaE1 in complex with the β-clamp and a mismatched DNA substrate shows how the primer strand is separated from the template strand and guided into the PHP-exonuclease active site during excision of a misincorporated nucleotide. Remarkably, the PHP active site shows a striking similarity to the active site of *E. coli* endonuclease IV, an enzyme involved in base excision repair, even though the two proteins do not appear to have a common evolutionary ancestor. Our work provides crucial insights into high-fidelity DNA replication in *M. tuberculosis* and reveals unique features of the PHP active site that can be exploited for development of new antibiotics to treat this major threat to global health.

## Results

**Overall structure of Mtb DnaE1.** The crystal structure of *M. tuberculosis* DnaE1 was determined by X-ray crystallography to 2.8 Å resolution. To promote crystallization, the C-terminal region was removed (residues 942–1187), analogous to the structure of *E. coli* DNA Pol IIIα[12]. The final structure (Fig. 1a, b)

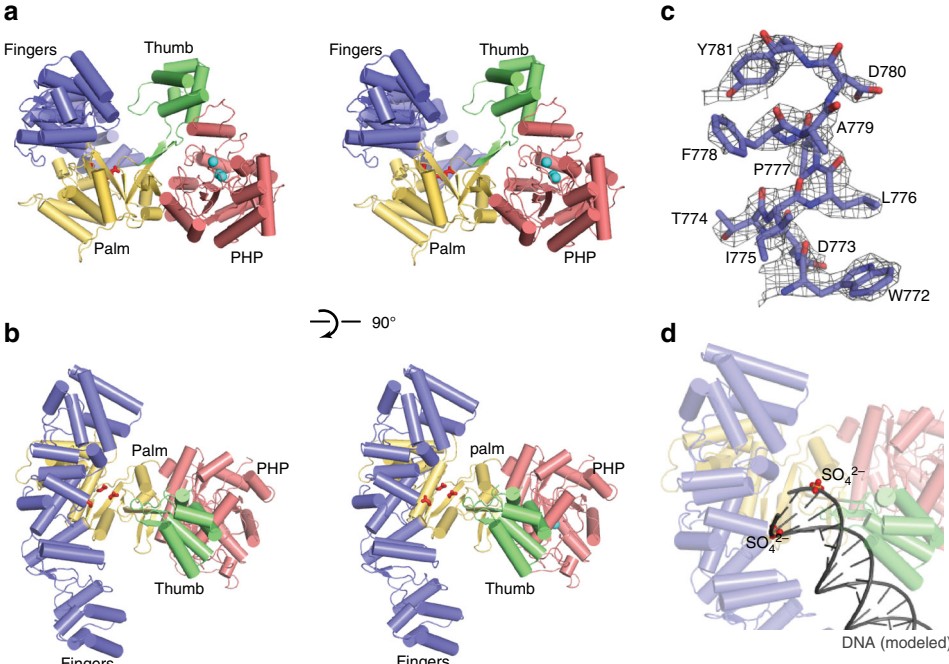

**Fig. 1** Crystal structure of Mtb DnaE1. **a** Stereo image of the polymerase with domains indicated in different *colors*. The catalytic residues of the polymerase are shown in *red sticks* and the three metals in the PHP-exonuclease site are shown in *blue spheres*. **b** Stereo image of the polymerase, rotated by 90° from the view in (**a**). **c** Example of the electron density showing clearly defined side chains. Electron density shown at 2.1σ in *gray*. A stereo image is shown in Supplementary Fig. 1a. **d** Modeling of dsDNA into the polymerase active site reveals an almost perfect overlap between the DNA backbone and two $SO_4^{2-}$ ions that are bound in the polymerase active site. See also Supplementary Fig. 1b for a comparison to other polymerases

**Table 1 Data collection and refinement statistics**

|  | DnaE1 | DnaE1 (SAD) | |
|---|---|---|---|
| *Data collection* | | | |
| Space group | H32 | H32 | |
| Cell dimensions | | | |
| $a, b, c$ (Å) | 256.0, 256.0, 187.3 | 255.9, 255.9, 187.4 | |
| $\alpha, \beta, \gamma$ (°) | 90.0, 90.0, 120.0 | 90.0, 90.0, 120.0 | |
| | | Low | High |
| Wavelength | 0.96864 | 1.2842 | 1.2815 |
| Resolution (Å) | 49.1–2.80 (2.88–2.80)[a] | 48.4–4.0 (4.47–4.0) | 48.4–4.0 (4.47–4.0) |
| $R_{merge}$ | 0.181 (1.295) | 0.089 (0.152) | 0.091 (0.136) |
| $I/\sigma I$ | 6.2 (1.2) | 8.9 (6.7) | 12.5 (8.5) |
| $CC_{1/2}$ | 0.975 (0.315) | 0.986 (0.972) | 0.989 (0.970) |
| Completeness (%) | 99.7 (99.7) | 68.2 (69.1) | 67.6 (68.4) |
| Redundancy | 3.4 (3.5) | 3.2 (3.2) | 3.3 (3.3) |
| *Refinement* | | | |
| Resolution (Å) | 2.80 | | |
| No. reflections | 57,575 | | |
| $R_{work}/R_{free}$ | 0.197/0.230 | | |
| No. atoms | | | |
| Protein | 7,127 | | |
| Ligand (Zn, $SO_4^{2-}$, HEPES) | 63 | | |
| Water | 81 | | |
| *B*-factors | | | |
| Protein | 47.4 | | |
| Zinc ions | 27.5 | | |
| Ligand ($SO_4^{2-}$, HEPES) | 90.3 | | |
| Water | 33.5 | | |
| R.m.s deviations | | | |
| Bond lengths (Å) | 0.0170 | | |
| Bond angles (°) | 2.0787 | | |

Data was cut at CC1/2 of 0.315
[a]Values in parentheses are for highest resolution shell

contains residues 1–933 and can be divided into four domains: the N-terminal PHP domain (residues 1–300), the thumb domain (residues 451–542), the palm domain (residues 301–450 and 543–591), and fingers domain (residues 592–933). The electron density map is of high quality, showing clearly defined side chains (Fig. 1c). The only exception is a loop in the thumb domain (residues 501–508) that is poorly defined in the density, most likely due to the absence of DNA. The final data and refinement statistics are given in Table 1. Overall, the structure of DnaE1 is similar to that of the α subunit of DNA polymerase III (PolIIIα) from *E. coli* and *Thermus aquaticus* (Taq)[12, 13] and to a lesser extend to the replicative polymerase PolC from *Geobacillus kaustophilus* (Gka)[14] (Supplementary Fig. 1b).

The polymerase active site, comprising aspartates 421, 423, and 587, is located at the bottom of the deep cleft between the thumb, palm, and fingers domain. In the structures of DNA-bound PolIIIα and PolC[11, 14, 15], this deep cleft binds the double-stranded DNA substrate, which can be modeled into the structure of DnaE1 without any major clashes (Fig. 1d and Supplementary Fig. 1b). Interestingly, in our structure two $SO_4^{2-}$ ions are bound at the bottom of the cleft and overlap with the phosphate atoms of the modeled DNA, indicating that the $SO_4^{2-}$ ions satisfy the propensity of the protein to bind the phosphates of the DNA backbone.

On the other side of the thumb domain, ~40 Å away from the polymerase active site, is the exonuclease active site (Fig. 1a). It is located in the center of the PHP domain and consists of nine conserved residues that coordinate three metals (Fig. 2a, b). The coordinating residues comprise five histidines, three aspartates/glutamates, and one cysteine. All residues are conserved in Taq and *Thermus thermophilus* PolIIIα, as well as *Bacillus subtilis* and *Thermus thermophilis* PolX (Supplementary Fig. 2), which like Mtb DnaE1 have been reported to have exonuclease activity[10, 13, 16, 17]. In contrast, in Gka PolC one of the residues is mutated[14], whereas in *E. coli* PolIIIα five of the nine residues are altered[9]. Consequently, neither of these two polymerases show exonuclease activity. Similarly, mutation of aspartate 23 or aspartate 226 in Mtb PHP domain renders the exonuclease inactive[6]. Unique to the Mtb PHP domain is a long, 29-residue loop (residues 77–105) adjacent to the PHP active site (Fig. 2c). This loop is conserved in the Actinobacteria, but less so in other bacteria (Supplementary Fig. 3). For example, a shorter, 18-residue loop is present in Taq PolIIIα (residues 76–93), whereas in *E. coli* Pol IIIα, the loop has been shorted to nine residues (residues 73–81). In Gka PolC, a DnaQ-like DEDD exonuclease is inserted into this loop, which sets the PolC polymerases apart from the other bacterial replicative DNA polymerases. In Mtb DnaE1, the long loop lines a narrow groove that forms a continuous cavity with the PHP-exonuclease active site, and has two $SO_4^{2-}$ ions bound (Fig. 2d).

**Cryo-EM analysis of the exonuclease state.** To reveal how the primer strand reaches the PHP-exonuclease active site, we used cryo-EM to determine the structure of DnaE1 bound to the β-clamp and a mismatched DNA substrate. Previously, we used a similar approach to determine the structures of the *E. coli* replicative DNA polymerase Pol III core bound to clamp and DNA[15, 18]. However, for the Mtb complex, we were unable to generate a 3D cryo-EM map due to a strong preferential orientation of the complex in the ice. Yet, some of the 2D class averages show clearly defined features of the polymerase and clamp as well as the DNA that appears to split in between the clamp and polymerase (Fig. 3a). Using the 2D class as a guide, we created a model of the complex, using the crystal structures of Mtb DnaE1 and the Mtb β-clamp[19] (Fig. 3b). First, DnaE1 and the clamp were positioned manually to fit the 2D class. Next, the DNA was modeled into the Y-shaped density between the clamp and PHP domain. As a control, we also created a model with the DNA in the polymerase mode and a model where the primer strand follows the narrow groove between the polymerase active site and PHP-exonuclease active site (Supplementary Fig. 4), but both of these models do not fit well with the 2D cryo-EM class. The Y-shaped DNA model shows that the primer strand is separated from the template strand by a short β-hairpin (residues 132–140) that acts as a wedge (Fig. 3c, d). The position of the primer strand fits well with hydrogen–deuterium exchange (HDX) experiments, which show that the long loop in the PHP domain is one of the regions most strongly affected by the presence of the DNA (Fig. 3c). Unfortunately, the HDX did not provide information about the short β-hairpin, as it was one of the 12 peptides that was not detected in the mass spectrometry analysis. Therefore, to further confirm the path of the DNA, we mutated five residues in the long loop or the β-hairpin: serine 100 to alanine (S100A), serine 102 to alanine (S102A), tyrosine 105 to phenylalanine (Y105F), lysine 136 to glutamine (K136Q), and tryptophan 137 to phenylalanine (W137F) (Fig. 3d). For comparison, we also used two mutants of metal-coordinating residues: aspartate 23 to asparagine (D23N) and aspartate 226 to asparagine (D226N) that both have lost exonuclease activity[6]. A third metal-coordinating mutant, glutamate 73 to glutamine (E73Q), resulted in loss of protein expression. All mutants were assayed for polymerase and

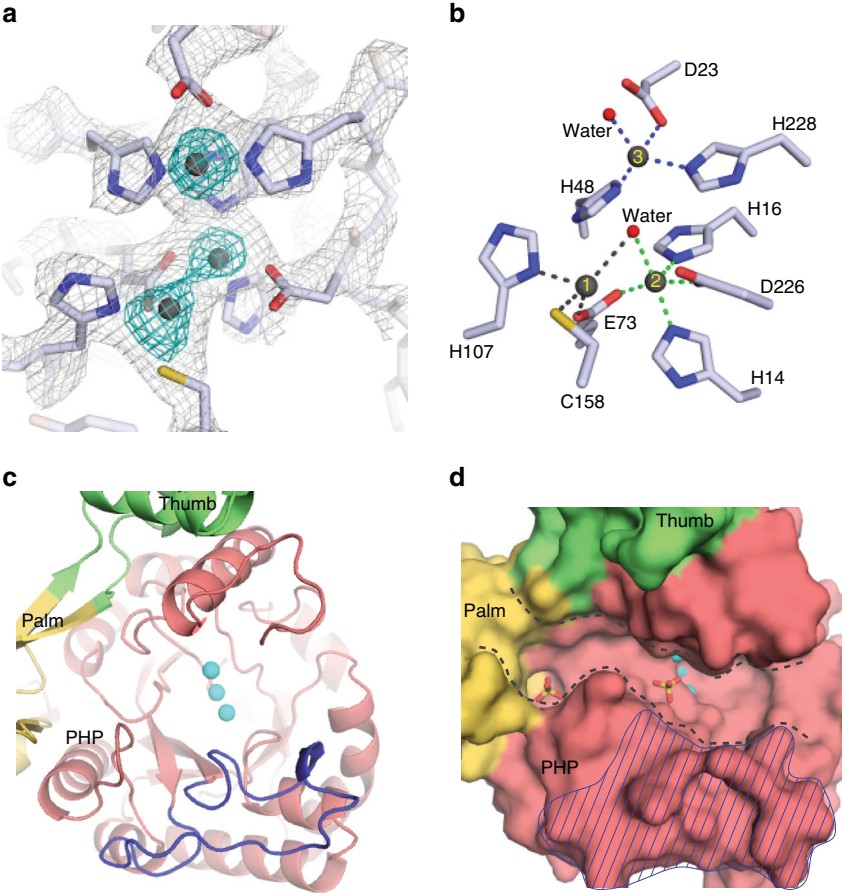

**Fig. 2** Structure of the PHP-exonuclease. **a** Close-up view of the PHP active site showing nine conserved residues that coordinate three metals. Electron density is shown at 2.0σ in *gray*, and at 8.2σ in *blue*. **b** Coordination of the three metal ions in the PHP active site. Metal 1 and metal 3 are coordinated in a tetrahedral geometry, whereas metal 2 is coordinated in a trigonal bipyramidal geometry. A water molecule is liganded between metal 1 and metal 2. **c** A uniquely long loop in the PHP domain, indicated in *dark blue*, forms a wall adjacent to the PHP active site. Metals are shown as *light blue spheres*. **d** A narrow channel, indicated with *dashed lines*, runs from the polymerase active site into the PHP-exonuclease site and contains two $SO_4^{2-}$ ions (shown in *stick*). The long loop is indicated with the *shaded area*. Metals are shown as *light blue spheres*

exonuclease activity using a real-time polymerase assay[20] using a matched and mismatched DNA substrate (Fig. 3e). In addition, their thermal stability was monitored using differential scanning fluorimetry (DSF), showing that all mutants apart from the metal-coordinating mutants (D23N and D226N) have similar melting temperatures to the wild-type (WT) protein (Fig. 3f). The five mutants in the DNA path show polymerase activity comparable to WT protein. On a mismatched DNA substrate, which requires exonuclease activity before the polymerase can initiate synthesis, both WT and S102A show a ~70% reduction of activity. The other DNA path mutants (S100A, Y105F, K136Q, and W137F) show an even stronger reduction of activity (~90%), whereas the metal-coordinating mutants D23N and D226N are completely devoid of the exonuclease activity. The latter also show reduced polymerase activity on a matched substrate, possibly caused by their decreased stability (Fig. 3f). The reduced exonuclease activity for the four mutants S100A, Y105F, K36Q, and W137F, strongly support the prediction that they are involved in DNA binding.

**The PHP-exonuclease has a trinuclear zinc center**. The catalytic center of the PHP domain has three metals bound (Fig. 2a, b), similar to structures of other PHP domains (Supplementary Fig. 2). The identity of these metals, however, is unclear as a variety of metals have been identified in different PHP domain structures, including zinc, iron, and manganese[21–24]. Therefore,

to identify the metals present in DnaE1, we performed an X-ray fluorescence scan on a crystal of DnaE1 (Fig. 4a). The scan shows two distinct peaks, one corresponding to the K-α emission line of the sulfurs (2308 eV) from the cysteines and methionines in the protein, and a second peak corresponding to the K-α emission line of zinc (8639 eV). No other metals were detected. Next, to exclude the possibility that the zinc metals were derived from one of the components of the crystallization conditions, we also measured the metal content of the purified protein using inductively coupled plasma optical emission spectrometry (ICP-OES). Here too, we find that zinc is the only metal present in the sample (Fig. 4b). Finally, to uniquely identify all three metals in the PHP active site as zinc, anomalous difference maps were calculated from two single-wavelength anomalous diffraction (SAD) data sets collected before (at 9655 eV) and after (at 9675 eV) the X-ray absorbance peak of zinc (at 9658 eV) (Fig. 4c, d). The presence of a strong anomalous difference map around the metals at the high energy, but not the low energy data set, unambiguously identifies all three metals as zinc. The presence of zinc in the PHP active site is consistent with previous work of the McHenry group that identified the PHP domain of *T. thermophilus* PolIIIα as a zinc-dependent enzyme[10], but not with the recent work of Gu et al.[25] that suggested that the DnaE1 exonuclease is a magnesium-dependent enzyme.

The zinc dependency of the PHP-exonuclease of DnaE1 is remarkable, as almost all high-fidelity DNA polymerases such as

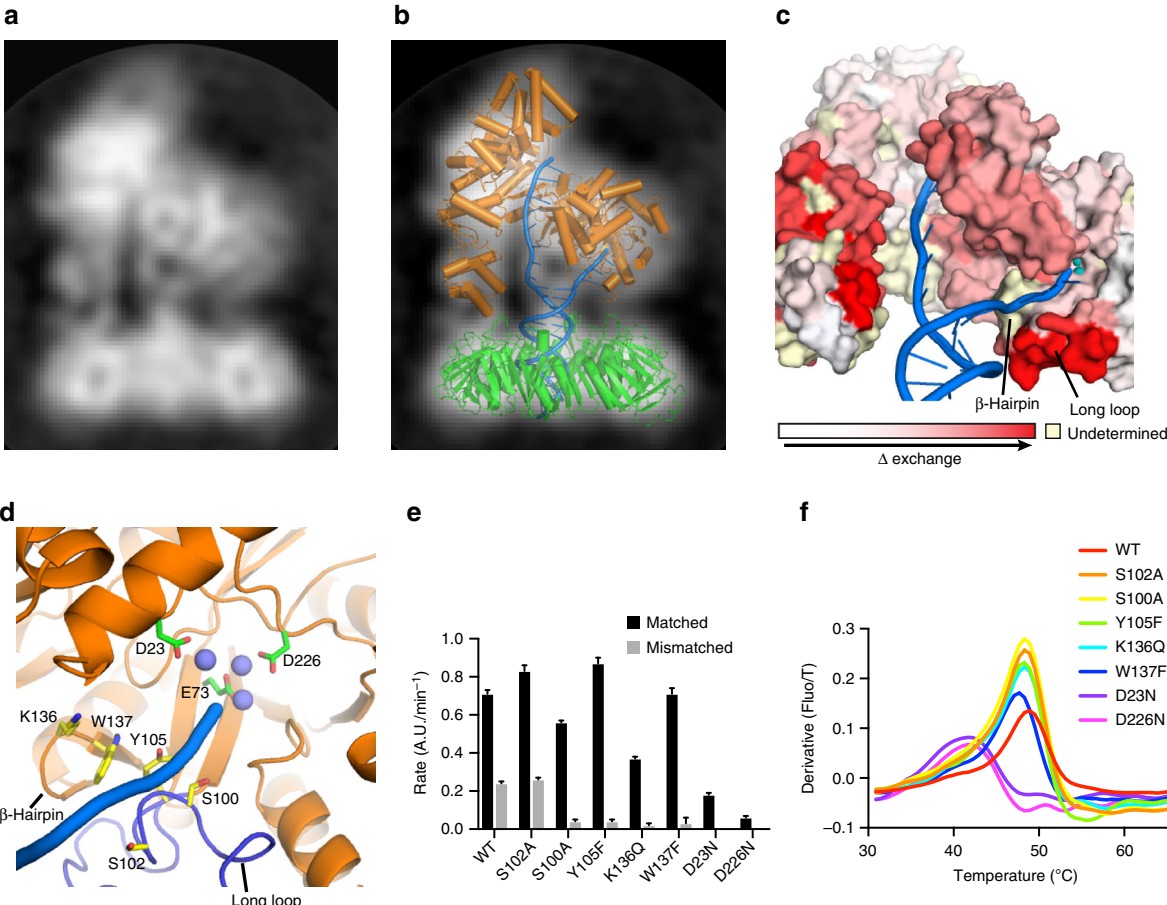

**Fig. 3** Entry path of the primer strand into the PHP-exonuclease. **a** 2D cryo-EM class of the Mtb DnaE1–clamp-DNA complex. **b** Fitting of the crystal structures of DnaE1 and β-clamp, and modeling of DNA into the 2D cryo-EM class. **c** Hydrogen–deuterium exchange experiments map the areas of the polymerase that are most affected by DNA binding, including the long loop in the PHP domain in contact with the modeled DNA. **d** Close up of the entry path and active site, with mutated residues indicated in *sticks*. The path of the modeled DNA is indicated by the *blue line*. **e** Primer extension rates of different DnaE1 mutants on a matched and mismatched DNA substrate calculated from three–five independent experiments. *Error bars* indicate standard deviation (Supplementary Fig. 5). **f** Differential scanning fluorimetry analysis of WT and mutant DnaE1

the *E. coli* DNA polymerase I, II, and III, as well as the eukaryotic replicative DNA polymerases pol δ and pol ε make use of DEDD-type exonuclease that uses magnesium for catalysis, similar to the polymerase itself. The polymerase and the canonical DEDD exonuclease both bind magnesium weakly, and require millimolar (mM) concentrations of magnesium for activity, which matches the intracellular concentration of Mg[26]. The PHP-exonuclease in contrast retains its catalytic metals throughout the protein purification procedure and does not require additional zinc for activity[6]. As the high concentration of magnesium required for the polymerase could potentially displace the zinc metals in the PHP domain, we wondered whether magnesium could affect the activity of the PHP-exonuclease. For this we monitored DNA polymerase activity using two DNA substrates, one with a matched primer, and one with a primer containing a mismatch at the 3′ end. Because of the high fidelity of the polymerase, the exonuclease first needs to remove the mismatched nucleotide before the primer can be extended. The DnaE1 polymerase activity shows a typical magnesium dependency that reaches maximum activity at 1–2 mM magnesium. In contrast, the exonuclease-dependent polymerase activity is reduced with increasing concentrations of magnesium, and is almost fully inhibited by 10–20 mM magnesium (Fig. 4e). This differs from the *E. coli* replicative exonuclease DnaQ (the ε subunit of the *E. coli* DNA polymerase III holoenzyme), which

reaches maximal activity at 10–20 mM magnesium (Fig. 4f). It is furthermore noteworthy that to compensate for the magnesium inhibition of the PHP-exonuclease, the DnaE1 polymerase already reaches optimal activity at 1 mM magnesium, whereas the *E. coli* polymerase only does so at 10 mM.

The inhibitory effect on the PHP-exonuclease is also obtained by other metals at comparable mM concentrations (Supplementary Fig. 6). Importantly though, only the intracellular concentration of magnesium is universally held at mM concentrations, whereas the intracellular concentration of the metals are $10^3$ to $10^6$-fold lower[26], indicating that the other metals will not be inhibitory in vivo.

**The PHP-exonuclease is strikingly similar to endonuclease IV**. To date, no mechanism for the PHP-exonuclease has been proposed. The only PHP enzyme for which a mechanism has been proposed is the L-histidinol phosphate phosphatase[22, 27] yet this PHP enzyme has a very different substrate than the PHP-exonuclease, and is therefore likely to have a different mechanism of substrate binding and hydrolysis. Unexpectedly, we found that the active site of the DnaE1 PHP domain shows a striking similarity to the active site of the *E. coli* endonuclease IV (Endo IV), an enzyme involved in base excision repair, whose mechanism has been studied in great detail[28–30]. Both enzymes

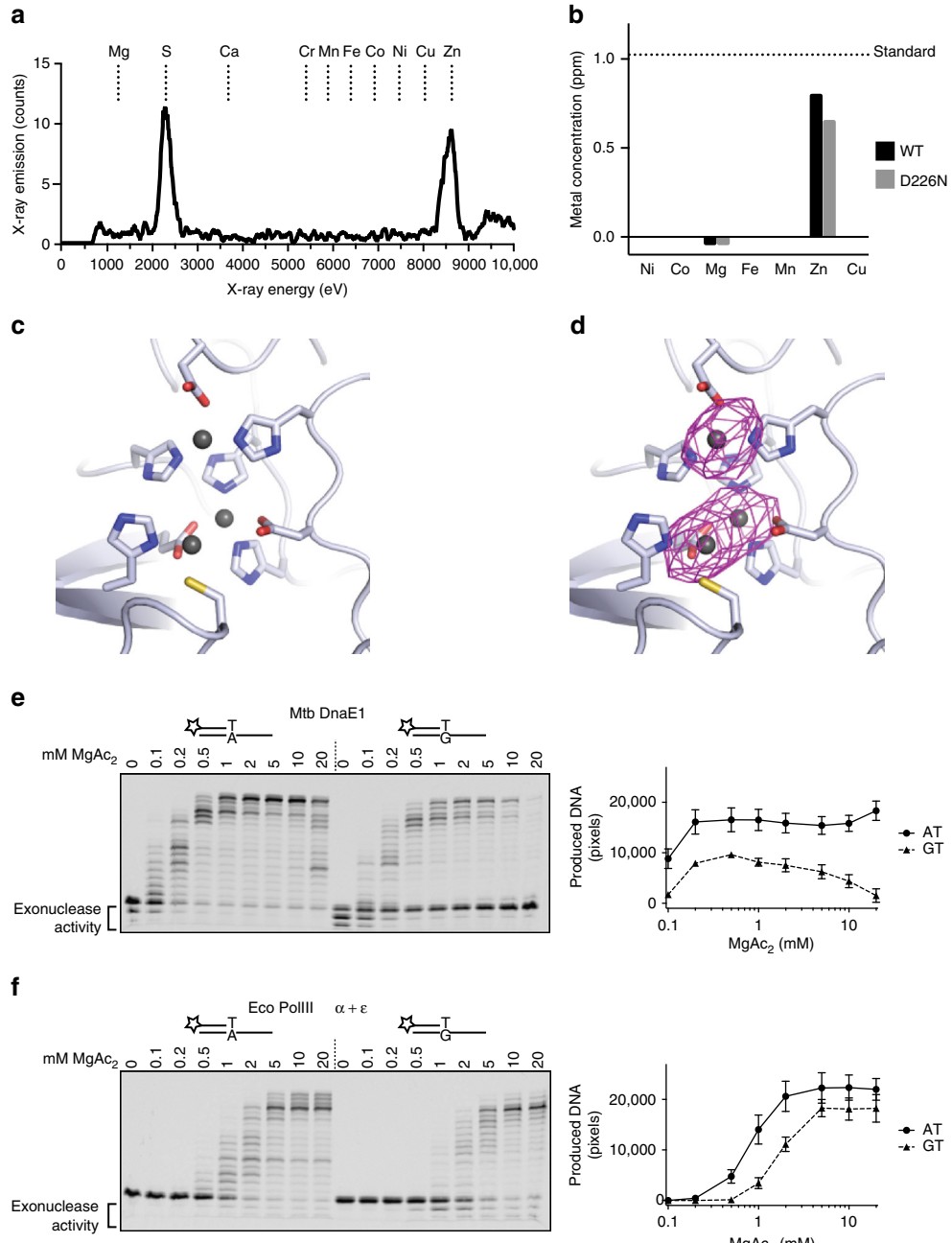

**Fig. 4** The PHP domain has a trinuclear zinc center. **a** X-ray emission scan of a DnaE1 crystal showing two distinct peaks at the sulfur and zinc emission energy. **b** ICP-OES analysis of purified DnaE1 protein confirms that only zinc is bound by the protein, in both wild-type and the catalytic mutant D226N. **c**, **d** Anomalous difference density maps before and after the zinc absorption edge uniquely identify all three metals in the PHP active as zinc. **e** Primer extension assays of a matched (AT) or mismatched (GT) DNA substrate reveals that the Mtb DnaE1 PHP-exonuclease is inhibited by magnesium. **f** In contrast, the *E. coli* exonuclease ε is activated by magnesium. The graphs on the *right* of the gels show the averaged amount of extended DNA from three independent experiments. *Error bars* indicate standard deviation. Uncropped gels are shown in Supplementary Fig. 8

have a trinuclear zinc center, which is coordinated by nine residues: histidines, aspartates/glutamates, or cysteine (Fig. 5a). A structural alignment of the nine residues reveals a high degree of similarity in their position (Fig. 5a). However, these residues originate from very different locations in their primary sequence (Fig. 5b) suggesting that the two enzymes do not share a common ancestor, but appear to be the result of convergent evolution. Furthermore, Endo IV consists of a single triosephosphate isomerase (TIM) barrel domain with eight strands and helices[31], whereas the PHP domains consists of seven strands and helices[7]. Despite this, the zinc-coordinating residues are almost identical

between the two enzymes, with the exception that aspartate 179 in Endo IV has been replaced by histidine 16 in the DnaE1 PHP domain, and histidine 69 in Endo IV has been replaced by cysteine 158 in DnaE1 PHP domain (Fig. 5c, d). Furthermore, the lower half of the active site appears to be "mirrored" between the two enzymes. Histidine 109 of Endo IV has moved to the opposite side of the active site in DnaE1 PHP (His 107) and has moved $Zn_1$ with it (Fig. 5c, d). The "mirroring" also extends to the two carboxylates (glutamate 261 and glutamate 145 in Endo IV, and glutamate 73 and aspartate 226 in DnaE1) that coordinate $Zn_1$ and $Zn_2$. In Endo IV, glutamate 261 coordinates $Zn_2$ and

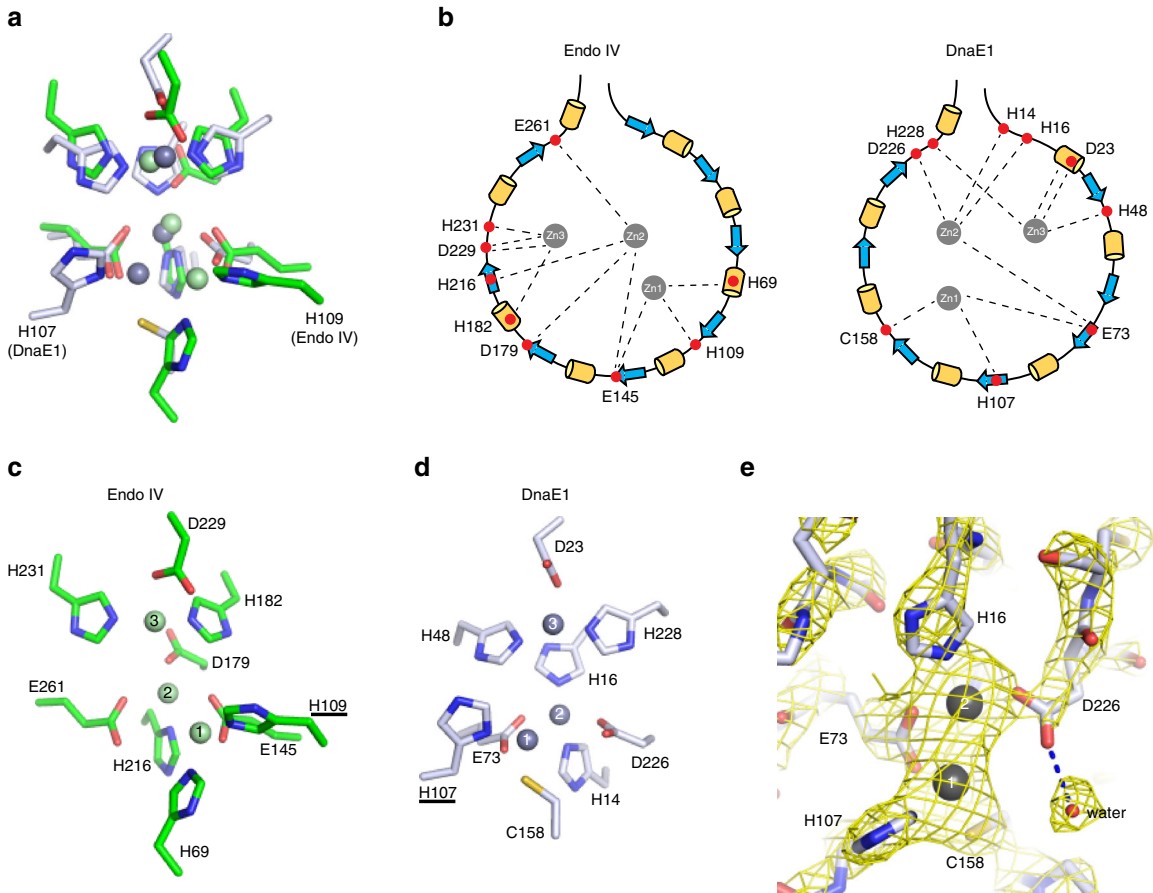

**Fig. 5** The PHP-exonuclease active site is analogous to endonuclease IV. **a** Overlay of the catalytic residues of DnaE1 (in *gray*) and *E. coli* Endo IV (in *green*). Note the mirrored position of Endo IV H109 and DnaE1 H107. **b** Schematic view of the positions of the residues that coordinate the zinc ions in Endo IV (*left*) and DnaE1 (*right*). The zinc-coordinating residues originate from entirely different locations, indicating the two enzymes do not have a common evolutionary origin. **c** Histidine 109 that ligands $Zn_1$ in Endo IV is positioned to the 'right' of the active site and pulls $Zn_1$ away from the center line. **d** In DnaE1, the analogous histidine (His 107) has moved to the other side of the active site, moving $Zn_1$ with it. **e** In DnaE1, glutamate 73 (E73) ligands both $Zn_1$ and $Zn_2$, whereas aspartate 226 (D226) only ligands $Zn_2$ and makes a hydrogen bond with a water molecule. Electron density show at $2.3\sigma$ in *yellow mesh*

a water molecule, which is deprotonated to a hydroxyl that performs the nucleophylic attack on the phosphodiester bond[30]. In DnaE1, this role appears to be taken up by aspartate 226, which makes and hydrogen bond to a water molecule (Fig. 5e). This is supported by mutation of the two carboxylates: a change of glutamate 73 into a glutamine resulted in loss of protein expression, consistent with a structural role in coordinating $Zn_1$ and $Zn_2$. A similar mutation of aspartate 226 to asparagine does not affect the protein expression or the metal content (Fig. 4b), but does results in a complete loss of exonuclease activity[6]. Hence, it is likely that in the Mtb PHP-exonuclease, aspartate 226 takes up the role of activating a water molecule for the nucleophilic attack on the phosphodiester bond of the bound DNA.

The structural correlation between Endo IV and the PHP-exonuclease is reminiscent of the eukaryotic nucleases APE1 and TDP2. APE1/TDP2 share mechanistic features with Endo IV[32] but are structurally distinct from Endo IV/PHP. APE1 is the functional equivalent of Endo IV in eukaryotic base excision repair that like Endo IV binds and distorts the double stranded DNA (dsDNA) to expose and incise adjacent to the abasic site[33]. TDP2 on the other hand is a single stranded DNA (ssDNA) exonuclease that resolves topoisomerase II-DNA adducts, and like the PHP-exonuclease binds a single stranded DNA substrate[34, 35]. Hence, in both the bacterial and eukaryotic nuclease pairs, Endo IV/PHP and APE1/TDP2, a similar mechanism is employed to cut the DNA, whereas substrate specificity is achieved through unique features in their DNA binding surface (Supplementary Fig. 7).

## Discussion

Tuberculosis is a major threat to global health, which is further aggravated by the frequent occurrence of drug-resistant strains[5]. Drug resistance in mycobacteria is caused by the point mutations in the DNA that are created de novo in each individual[36]. Therefore, to understand the mechanisms that drive drug resistance, it is essential to understand the mechanisms that control the mutation rate in this bacterium. As mycobacteria lack DNA mismatch repair, the only process that can ensure low levels of mutation during DNA replication is the replicative DNA polymerase and the associated proofreading exonuclease. *M. tuberculosis* does not use the canonical DnaQ-like DEDD exonuclease that is found in the well characterized *E. coli* replication machinery, as well as in the eukaryotic replicative DNA polymerase Pol δ and Pol ε. Instead, the Mtb replicative DNA polymerase DnaE1 uses a PHP-exonuclease that is intrinsic to the protein itself. It was only recently that the full role of the PHP domain in DNA replication was revealed[6]. Consequently, the mechanism by which it operates has remained undetermined.

The crystal structure of Mtb DnaE1 presented in this work provides a detailed insight into the actions of the PHP-exonuclease. We show that the exonuclease has a trinuclear

zinc center and that it has a striking similarity to *E. coli* endonuclease IV, an enzyme involved in base excision repair. The use of zinc in a nuclease is unusual, as the large majority of nucleases use magnesium as the catalytic metal[37]. The presence of zinc in the PHP-exonuclease is remarkable and allows us to draw two possible conclusions. First, the presence of other metals such as $Mg^{2+}$, one of the most abundant metals in living organisms[38], may have an inhibitory effect on the actions of the PHP-exonuclease. Indeed, we show that the millimolar concentrations of $Mg^{2+}$ that are used by DNA polymerases inhibit the PHP-exonuclease activity. Accordingly, changes in concentration of magnesium could temporarily inhibit the proofreading activity of the polymerase resulting in increased mutagenesis. Other mechanisms that increase mutation rates have been linked to drug resistance, such as inactivation of DNA mismatch repair[39], and increased expression of error prone DNA polymerases such as *E. coli* DinB (Pol IV)[40], and the Mtb DnaE2[4]. It is therefore tempting to speculate that inactivation of the DnaE1 PHP-exonuclease through increased $Mg^{2+}$ concentrations may provide an alternative way to temporarily increase mutation rates to enable rapid evolution and select for drug resistance.

The second conclusion stemming from the zinc-dependent exonuclease is that in certain environments it could be unfavorable to have an exonuclease that is inhibited by $Mg^{2+}$, one of the most abundant divalent metals in biology. Indeed, we have found that in at least three independent events in evolution the PHP-exonuclease has been replaced by the $Mg^{2+}$-dependent DEDD exonuclease: as an insertion into the PHP domain in firmicutes, as an N-terminal fusion in bacteroidetes, and as an exonuclease in trans in the α, β, and γ proteobacteria[6, 41]. This, however, does not exclude that other additional factors may have contributed to the selection of a magnesium-dependent exonuclease over the zinc-dependent PHP-exonuclease.

Finally, our structure also reveals a narrow groove that together with the PHP active site forms a unique cavity that appears suitable for structure-based design of inhibitors to the PHP-exonuclease. Increasingly, DNA replication, and in particular, the replicative DNA polymerase, is being recognized as an attractive target for novel antibiotics[42–44]. Likewise, we have shown that inactivation of the PHP-exonuclease renders the polymerase sensitive to nucleotide analogs[6] that are used in antiviral therapy including in treatment of patients with HIV[45]. This is of particular interest, as many patients with TB are also co-infected with HIV[46] and would benefit greatly from a single treatment that targets both diseases simultaneously. The PHP-exonuclease is an especially attractive target as this domain is not found in humans, making the likelihood of cross reactivity small. The crystal structure of Mtb DnaE1 presented in this work and the analysis of its PHP-exonuclease is therefore a valuable tool in the development of novel antibiotics that target DNA replication in *M. tuberculosis*.

## Methods

**Protein purification and crystallization.** WT and mutants *M. tuberculosis* DnaE1 as well as the short version of DnaE1 (residues 1–941) were expressed in *Mycobacterium smegmatis* mc²1457 cells, kindly provided by W. Jacobs Jr (Albert Einstein College of Medicine). The proteins were purified using nickel-affinity, anion-exchange, and gel filtration columns, and His tags were cleaved with HRV 3C protease. All purification steps were carried out in 50 mM HEPES, pH 7.5, 0.1–1 M NaCl and 2 mM DTT, and stored at −80 °C in 50 mM HEPES pH 7.5, 150 mM NaCl, and 2 mM DTT. DnaE1 crystals were obtained by hanging drop vapor diffusion method from protein at 5 mg/ml and mother liquor solution containing 1.2 M Li₂SO₄, 0.1 M HEPES pH 7.5. Crystals were transferred to well solution including 25% glycerol before flash freezing in liquid nitrogen.

**Diffraction data collection and structure determination.** Diffraction data were collected at 100 K at beamline I24 at the Diamond Light Source. A data set to 2.8 Å resolution was processed in XDS[47] and scaled using Aimless[48]. DnaE1 crystal

belonged to the space group H32 with the following unit-cell dimensions: $a = 255$ Å, $b = 255$ Å, $c = 187$ Å. Data were cut at $CC_{1/2}$ value of 0.315[48]. The structure of Mtb DnaE1 was solved by molecular replacement in Molrep[49] using the structure of the α subunit of *E. coli* polymerase III as a model. The model was built in Coot[50] and refined in Refmac5[51]. The final $R_{work}$ and $R_{free}$ of the final model are 19.6 and 23%, respectively, with 93.7% of the residues in the preferred regions, and 6.7% in the allowed regions of the Ramachandran plot. The final structure does not contain residues 501–508, corresponding to the flexible loop in the thumb domain. refinement statistics are given in Table 1. All figures were made using PyMOL[52]. Electron density maps shown in Figs. 1c, 2a, and 5e are 2mFo-DFc composite omit maps calculated in PHENIX[53], and shown using carve levels of 2.4, 2, and 2.5, respectively.

**Cryo-EM data collection and data processing.** Cryo-EM data collection and data analysis were performed in a similar manner to the *E. coli* PolIIIα-exonuclease–clamp-DNA complex[15]. In brief, Mtb DnaE1 and β-clamp were mixed in equimolar ratio and purified by gel filtration. To the peak fraction, a tenfold excess of a double-stranded DNA substrate with a mismatch at the 3′ of the primer was added before preparation of the cryo-EM sample grid (template strand: 5′-CCCCTCAGGAGTCCTTCGTCCTAGTACTACTCC, primer strand: 5′-GGA GTAGTACTAGGACGAAGGACGsT (where lower case "s" indicate a phosphothioate bond). Data were collected using a Titan Krios electron microscope (FEI) operated at 300 kV equipped with a K2 Summit direct electron detector (Gatan) mounted after a Gatan Imaging Filter (GIF) using a 20 eV slit to remove inelastic scattered electrons. 20 frame image stacks were collected in electron counting mode using a flux of 2 e/Å²/s and a total dose of 40 e/Å². Frames were aligned and averaged using MOTIONCORR[54]. Contrast transfer function parameters were calculated using Gctf[55]. All subsequent particle picking and data processing was done in a pre-release version of Relion-2[56]. The final data set contained 69,935 particles, of which 4338 particles (6.2%) were in the 2D class average used for the modeling of the DnaE1–clamp-DNA complex. For the modeling, the crystal structures of Mtb DnaE1 and Mtb clamp were manually positioned in PyMOL[52] to fit with the 2D class average shown in Fig. 3b. A dsDNA model was created and modified manually in Coot[50].

**Real-time primer extension assays.** Polymerase and exonuclease activity were followed by a real-time fluorescent primer extension assay[20]. Reactions were performed in 50 mM Hepes pH 7.5, 50 mM potassium glutamate, 10 μM dGTP, 2 mM MgSO₄, 2 mM DTT, and 6 mg/ml BSA, using 20 nM labeled DNA and 100 nM unlabeled DNA (template strand: 5′-6-FAM-CCCCCCCCCCGCACCTAAAGTTG GGAGTCCTTCGTCCTA, primer strand: 5′-TAGGACGAAGGACTCCCAACT TTAGGTGC). Reactions were initiated by addition of 30 nM DnaE1 (final concentration) and measured in a 384-well plate using a BMG Labtech Pherastar FS plate reader during 15 min with 5 s intervals at 22 °C. Data were normalized and analyzed in GraphPad Prism. Three to five independently measured curves were used to fit a single exponential decay equation, as shown by a *red curve* in Supplementary Fig. 5.

**Thermo stability analysis of WT and mutant DnaE1.** Protein melting temperatures were determined via differential scanning fluorimetry using a Corbett Rotor-Gene 6000 real-time qPCR thermocycler. Overall, 0.5 μM DnaE1 WT or mutants were mixed with 10 × SYPRO orange (Sigma) in 20 mM Bis-Tris pH 6.5, 50 mM K-Glu, 0.5 mM TCEP. Excitation and emission filters were set to 460 and 510 nm, respectively. Heating from 30 to 85 °C, a constant heating rate of 0.6° per step was applied. The melting temperature ($T_m$) of the protein (0.5 μM) was determined from the inflection point of the first derivative of the melting curve. All samples were measured in triplicate.

**Identification of zinc sites by SAD.** Single-wavelength anomalous diffraction (SAD) data were collected at 100 K at beamline I24 at Diamond Light Source. A fluorescence scan over a DnaE1 crystal allowed us to identify the presence of Zn in the protein. Two data sets were collected before (9655 eV) and after (9675 eV) the absorption peak for zinc. To reduce radiation damage, the beam was attenuated to a diffraction limit to ~4 Å. To further avoid radiation damage, the rod-shaped crystal was translated after each 20° and data were collected using inverse beam. Each wedge of the diffraction data was processed with XDS independently and scaled and merged in Aimless setting the resolution limit to 4 Å. Anomalous difference maps were calculated using PHENIX[53].

**Metal content determination by ICP-OES.** A protein solution of DnaE1 was diluted to 10 μM in 0.1 M HEPES pH 7.5, 150 mM NaCl, 2 mM DTT. A standard solution containing 1 ppm of different divalent cations (Ni²⁺, Co²⁺, Mg²⁺, Fe²⁺, Mn²⁺, Zn²⁺, Cu²⁺) was used as a reference. Samples were analyzed on a Perkin Elmer ICP-OES chemical analyzer at the Department of Geography of the University of Cambridge, United Kingdom.

**Gel based DNA polymerase and exonuclease activity assays.** DNA oligos for activity assays were purchased from Integrated DNA Technologies, UK. The

labeled DNA primer oligo [6FAM]-5′-GAGTCCTTCGTCCTT-3′ was purified on a denaturing 20% acrylamide gel and used for exonuclease assays. For the polymerase assays, the purified primer was annealed with the unlabeled DNA template oligos 3′-CTCAGGAAGCAGGAATTTTGCTGTCTGTGAA-5′ or 3′-CTCAGGAAGCAGGAGTTTTGCTGTCTGTGAA-5′ to yield matched or mismatched substrates, respectively (underlined nucleotides indicate position opposite 3′ terminal nucleotide of the primer strand). dsDNA substrates were stored at 1.25 µM in 10 mM Tris-HCl pH 8 and 1 mM EDTA at 20 °C. DNA polymerase activity assays were performed in 50 mM HEPES pH 7.5, 50 mM potassium glutamate, 6 mg/ml BSA, 2 mM DTT and varying magnesium acetate concentrations as indicated. Primer extension was carried out at 20 °C with 6 nM purified protein (or protein complex) and 100 nM DNA for 5 min in the presence of 100 µM dNTP. Reactions were quenched in 50 mM EDTA pH 7.4, separated on a denaturing 20% acrylamide gel and imaged with a Typhoon Imager (GE Healthcare, UK). Three independent experiments were performed for quantification of extended primer strand. Gel band intensities were quantified using ImageJ software[57]. Metal competition assays reactions contained 2 mM Mg, and increasing concentrations (0–20 mM) of ZnCl$_2$, MnCl$_2$, NiCl$_2$, or CoCl$_2$.

**HDX sample preparation and experiment.** Deuterium exchange reactions of DnaE1 with and without DNA were initiated by diluting the protein at 25 µM concentration to give a final D$_2$O percentage of 95.3. All experiments were performed in triplicate and contained four time points: 3 s on ice, 3 s at 23 °C, 30 s at 23 °C, and 300 s at 23 °C. Samples were quenched by the addition of chilled 2 M guanidinium hydrochloride in 2.4% v/v formic acid and immediately frozen in liquid nitrogen. Samples were stored at −80 °C prior to analysis. The quenched protein samples were thawed and digested by pepsin followed by reversed phase HPLC separation. Peptides were detected on a SYNAPT G2-Si HDMS mass spectrometer (Waters, UK) acquiring over a m/z of 300 to 2000, with the standard electrospray ionization source and lock mass calibration using [Glu1]-fibrino peptide B (50 fmol/µl). The mass spectrometer was operated at a spray voltage of 2.6 kV and a source temperature of 80 °C. Spectra were collected in positive ion mode. Elevated energy mass spectrometry (MS$^e$)[58] was used for peptide identification, using an identical gradient of increasing acetonitrile in 0.1% v/v formic acid over 12 min. All resulting MS$^e$ data were analyzed using Protein Lynx Global Server software (Waters, UK) with an MS tolerance of 5 ppm, and mass analysis of the peptide centroids performed using DynamX software (Waters, UK). Only peptides with a score >6.4 were considered. Deuterium incorporation was not corrected for back-exchange and therefore represents relative, rather than absolute changes in deuterium levels. All time points in this study were prepared at the same time and individual time points were acquired on the mass spectrometer on the same day.

**Data availability.** The coordinates and the structure factors have been deposited in the Protein Data Bank under accession code 5LEW. Other relevant data are available from the corresponding author upon reasonable request.

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

## Acknowledgements

We thank Diamond Light Source for access to beamline I24. We also thank Garib Murshudov for help and suggestions, and Sarah S. Fortune and Jeremy Rock for critical reading of the manuscript. We thank Rafael Fernandez-Leiro for help with cryo-EM data collection and analysis. This work was supported by the UK Medical Research Council through grant U105197143.

## Author contributions

S.B.-M. and M.H.L. designed experiments. S.B.-M. purified and crystallized the protein. S.B.-M. and M.H.L. collected X-ray diffraction data, determined and refined the structure. S.B.-M. and M.H.L. collected and analyzed cryo-EM data. U.F.L. performed DNA polymerase extension assays. A.-M.M.v.R. purified protein and performed real-time primer extension and thermofluor assays, and S.L.M. and J.M.S performed hydrogen–deuterium exchange experiments and analyzed the data. M.H.L. and S.B.-M. prepared the manuscript with input from all authors.

## Additional information

**Competing interests:** The authors declare no competing financial interests.

