## [Peer Review file · Nature Communications]

Reviewers' comments:

Reviewer #1 (Remarks to the Author):

The manuscript presenting the crystal structure and analysis of the *Mycobacterium tuberculosis* DNA polymerase is quite interesting and thought provoking. One would have thought that the exonuclease in the polymerase is conserved across the microbial kingdom, but this is not the case. The finding of an EndoIV-like three Zn nuclease adds to the surprise and significance. EndoIV is an abasic DNA damage endonuclease, and there is not a direct connection between the damage endonuclease and a mismatch exonuclease. The crystallography is done superbly, including the detective work on the zinc element. This work, particularly due to its combination of strong science and the high health relevance of this microbe associated with tuberculosis, is of general interest to readers of *Nature Communications*.

The authors should consider the following points.

Points.

1. The authors mention that they used PolIIIalpha and PolC to help determine the dsDNA path. Is the path identical, if the structures are overlaid or is some shifting required?
2. It is interesting that EndoIV acts primarily on dsDNA and the PHP domain acts on ssDNA. It is highly reminiscent of Ape1 and TDP2, with a similar narrowing of the active site. The readers may benefit if the related mechanism is considered.
3. It is too bad that this particular structure does not actually contain DNA but this does not undercut the novelty and importance of the main points. The mechanism is truly intriguing. I wonder whether a portion of the dsDNA would not indeed fit into the narrow groove, with the last nt being required to be unpaired. It would provide a mechanism to preferentially incise mismatches, whose single nucleotide could be easily disrupted from the duplex DNA. This brings the question whether the PHP exonuclease can work processively on long overhangs (which would be consistent with a long ss channel), processively on duplex DNA (which is suggested from the long ss channel), or (my guess) one incision on a single mismatched nt at the end of duplex DNA.
4. Beyond the zinc coordination, the residue required for catalysis was identified in EndoIV. Is it conserved in residue type and location in the PHP domain?
5. The mirror relationship of the catalytic sites in PHP and endoIV raises the question whether or not this is more general with the AP-endonuclease site so readers may wonder how PHP compares to the human AP-endonuclease APE1 (i.e. Tsutakawa SE et al., Conserved structural chemistry for incision activity in structurally non-homologous apurinic/apyrimidinic endonuclease APE1 and endonuclease IV DNA repair enzymes. *J Biol Chem.* 2013 Mar 22;288(12):8445-55). Does the mechanism match that of EndoIV and/or APE1? Also one presumes that any overall catalytic site similarities would not limit structure-based inhibitor design against the PHP versus human APE1? One of the zincs could be replaced by Mn in EndoIV. Can anything be learned from comparing this zinc site to that in PHP?
6. The perspectives in Figure 3 a and b are difficult to follow, even with the zinc ions as a guide. Is the long loop visible in Figure 1 and can be labeled? Is the long loop conserved in sequence?
7. The authors mention the high sequence fidelity of this polymerase over the others. Is there something special about having a zinc exonuclease that may provide greater efficiency? Is there precedence for EndoIV to work on mismatches?
8. Electron density maps are provided in some figures. These maps should be simulated annealing omit maps (and described as such) and the type of map (e.g. 2mfofc), sigma level, and carving

should be noted.

9. The number of ligands in the crystal structure is surprising. Assuming three zincs, what are the other ones? Perhaps the B-factors for the zincs, as they are the most central, should be provided in their own right. What's the evidence for the other ligands, as the resolution is only 2.8 ?

Reviewer #2 (Remarks to the Author):

This study by Banos-Mateos et al. describes the molecular details of a crystal structure for the majority of the DnaE1 high-fidelity replicase from *M. tuberculosis*, detailing the physical relationship between the polymerase and exonuclease domains. This story follows on from their previous study (Rock et al., 2015) in which they reported that the PHP domain acts as the proofreading exonuclease subunit for this polymerase, enabling it to perform high-fidelity synthesis.

The structural studies reveal that this replicase has a similar architecture to that of the previously reported and closely-related replication polymerases from other bacteria, such as PolC and PolII. Using sequence analysis and HDX, they predict a putative ss DNA binding channel that runs between the active sites of the polymerase and PHP subunits and propose its possible role in directing DNA during synthesis for the removal of misincorporated bases by the PHP subunit. They subsequently describe the zinc binding properties of this domain, including the preferential binding of zinc ions to the exonuclease and the inhibition of its activity by an excess of magnesium ions. They then show that there is a significant similarity between the active sites of this PHP domain and EndoIV. Finally, based on these collective insights, they propose a zinc-dependent mechanism for this PHP exonuclease.

Major points

1. In figure 2, the structure of the PHP domain active site, including bound metal ions, is described. The structure of other PHPs are known, such as those from bacterial Pol Xs, and therefore the authors should compare these different structures to the Mt PHP domain, even if it is a supplementary figure, and also reference these key papers as they were among the first to describe this fold having exonuclease activity.
2. Although they use structure, sequence and HDX analysis to identify a putative ss DNA binding channel. The data presented in figure 3 is anecdotal and circumstantial and only weakly supports their model. In particular, the sequence data in Figure 3a is very confusing and could mean a range of things so it is not particularly persuasive. The authors need to provide much more functional biochemical / biophysical evidence to validate that this channel actually binds such substrates. For example, mutate the residues that line the channel and study the consequences on DNA binding and catalysis. Use crosslinking to show DNA binding in this region. Use FRET probes on DNA to demonstrate binding in the channel. Ultimately, these models can only be properly validated when they have a structure with ss DNA bound, this key piece of the puzzle is currently lacking from this study.
3. While it is clear from their studies that zinc ions preferentially bind in the PHP active site, they have not determined the KD of binding for this or various other competing metal ions, which could be achieved by doing competition assays using a range of methods. This is essential because in figure 4e & f they show that magnesium ions inhibit the exonuclease activity. Although this is interesting, the next step is to show how this occurs and why, they have not investigated this finding in any significant way.

4. The demonstration that the active site of PHP shows a "mirroring" of the zinc ions binding sites of EndoIV is interesting and based on the DNA-bound structures of this nuclease they propose a catalytic mechanism for this PHP. Although it is legitimate to do this, such models then need to be validated using experimental biochemical studies, e.g. classical kinetic approaches combined with site-directed mutagenesis, to evaluate the proposed nucleolytic cycle. It is essential that such studies are undertaken to back up the model presented in figure 7.

Minor Points

5. In figure 1, the residue numbers label should be labelled in the electron density maps, although I'm not sure it is really needed and this could go into the supplementary section.

6. In figure 2, they refer to either "Zinc" or "Metal" to describe the bound ions, Given they have not yet shown it is zinc, demonstrating this later in the paper, maybe they should use the term "Metal" ?

Point-by-point response to the reviewers

Reviewers points are shown in red font.

Our answers are shown in blue font.

Reviewer #1 (Remarks to the Author):

The manuscript presenting the crystal structure and analysis of the Mycobacterium tuberculosis DNA polymerase is quite interesting and thought provoking. One would have thought that the exonuclease in the polymerase is conserved across the microbial kingdom, but this is not the case. The finding of an EndoIV-like three Zn nuclease adds to the surprise and significance. EndoIV is an abasic DNA damage endonuclease, and there is not a direct connection between the damage endonuclease and an mismatch exonuclease. The crystallography is done superbly, including the detective work on the zinc element. This work, particularly due to its combination of strong science and the high health relevance of this microbe associated with tuberculosis, is of general interest to readers of Nature Communications.

The authors should consider the following points.

1. The authors mention that they used PolIIIalpha and PolC to help determine the dsDNA path. **Is the path identical, if the structures are overlaid or is some shifting required?**

Indeed, the path is identical. To illustrate this, we show the superposition of the three structures below (Reviewer1-Fig. 1). The cryo-EM structure of *E. coli* Pol III α -clamp-exo-DNA complex (pdb code: 5FKW) and the crystal structure *G. kaustophilus* (3F2B) were superimposed onto Mtb DnaE1 palm and fingers domain (residues 307-933). We have also included this figure in Supplementary Fig. 1.

Reviewer1-Figure 1. Superposition of *E. coli* PolIII α -DNA (green), *G. kaustophilus* PolC-DNA (orange), and *M. tuberculosis* DnaE1 (blue). A double stranded DNA molecule (black) was modeled into the DnaE1 structure using the other DNA molecules as a guide.

2. It is interesting that EndoIV acts primarily on dsDNA and the PHP domain acts on ssDNA. It is highly reminiscent of Ape1 and TDP2, with a similar narrowing of the active site. The readers may benefit if the related mechanism is considered.

In the original version of the manuscript we decided not include APE1/TDP2 as these enzymes are structurally distinct from EndoIV/PHP and have a different active site. Yet, we agree with the reviewer that there are indeed many interesting similarities between EndoIV/PHP and APE1/TDP2. We have therefore included a paragraph at the end of the section "*The PHP-exonuclease shows a striking similarity to Endonuclease IV*" (page 12, lines 3-12) that discusses the similarities. We have also included a structural comparison between the different enzymes in Supplementary Fig. 7.

3. It is too bad that this particular structure does not actually contain DNA but this does not undercut the novelty and importance of the main points. The mechanism is truly intriguing. I wonder whether a portion of the dsDNA would not indeed fit into the narrow groove, with the last nt being required to be unpaired. It would provide a mechanism to preferentially incise mismatches, whose single nucleotide could be easily disrupted from the duplex DNA.

It seems unlikely that dsDNA can bind in the narrow groove. Modeling of a dsDNA molecule into the groove (with the 3' of the primer strand in the PHP active site) results in significant clashes with the protein (see Reviewer1-Fig2 below). Furthermore, in order for the primer strand to reach the PHP active site, the dsDNA molecule would need to rotate by $\sim 105^\circ$. This in turn would cause additional clashes in other parts of the protein.

Reviewer1-Figure 2. Modeling of a dsDNA molecule into the narrow groove leading into the PHP active site. The protein is shown in blue surface, the primer strand in red, and the template strand in black. In this position the DNA has multiple clashes with the protein and is rotated by ~ 105 degrees from its original position. The yellow lines indicate the central axis of the dsDNA and the angle between the two positions.

This brings the question whether the PHP-exonuclease can work processively on long overhangs (which would be consistent with a long ss channel), processively on duplex DNA (which is suggested from the long ss channel), or (my guess) one incision on a single mismatched nt at the end of duplex DNA.

This is correct. Previously, we have shown that the PHP-exonuclease can indeed remove multiple nucleotides from a ssDNA substrate, whereas on a dsDNA substrate only the mismatched nucleotide is removed. (see Fig 1f & h in: Rock, Nat. Genetics, 2015)

Rock JM, Lang UF, Chase MR, Ford CB, Gerrick ER, Gawande R, et al. DNA replication fidelity in Mycobacterium tuberculosis is mediated by an ancestral prokaryotic proofreader. Nat Genet. 2015, 47 pp677–81.

4. Beyond the zinc coordination, the residue required for catalysis was identified in EndoIV. Is it conserved in residue type and location in the PHP domain?

As we propose in the section "Mechanism of the PHP-exonuclease" we believe that Asp226 takes up the role of the catalytic residue that deprotonates a water molecule for the nucleophilic attack on the phosphodiester bond. We have now added additional mutagenesis data that are consistent with a catalytic role for Asp226 (page 11, lines 6-11).

5. The mirror relationship of the catalytic sites in PHP and endoIV raises the question whether or not this is more general with the AP-endonuclease site so readers may wonder how PHP compares to the human AP-endonuclease APE1 (i.e. Tsutakawa SE et al., Conserved structural chemistry for incision activity in structurally non-homologous apurinic/apyrimidinic endonuclease APE1 and endonuclease IV DNA repair enzymes. J Biol Chem. 2013 Mar 22;288(12):8445-55). Does the mechanism match that of EndoIV and/or APE1?

We believe that the mechanism of the PHP-exonuclease is similar to that of EndoIV, but not to that of APE1. This is discussed in the sections "*The PHP-exonuclease shows a striking similarity to Endonuclease IV*" and "*Mechanism of the PHP-exonuclease*".

Also one presumes that any overall catalytic site similarities would not limit structure-based inhibitor design against the PHP versus human APE1?

Indeed, the conserved narrow groove and cavity adjacent to the PHP active site appear unique to the PHP-exonuclease. We are currently actively screening for specific inhibitors to the Mtb PHP-exonuclease.

One of the zincs could be replaced by Mn in EndoIV. Can anything be learned from comparing this zinc site to that in PHP?

Unfortunately, the resolution of our structure (2.8 Å) only allows us to do a simple comparison of the two active sites. It is clear that the two active sites are very similar, and that the position of the metal binding residues closely overlap (see Fig 5). There may be small differences between the two active sites that can influence metal preferences but these details cannot be determined at the current resolution.

6. The perspectives in Figure 3 a and b are difficult to follow, even with the zinc ions as a guide. **Is the long loop visible in Figure 1 and can be labeled?**
We have now amended the figure to be more clear.

Is the long loop conserved in sequence?

The loop is indeed conserved within the phylum of Actinobacteria, where the loop length is 30-35 residues. Outside this phylum, in δ Proteobacteria and Firmicutes the loop is shortened to 15-18 residues and shows little conservation between species. We have now added a section to the text (page 6, lines 4-5), and show a sequence alignment of the long loop in Supplementary Fig. 3.

7. The authors mention the high sequence fidelity of this polymerase over the others. **Is there something special about having a zinc exonuclease that may provide greater efficiency?**

The fidelity of the polymerase is determined by three processes: the insertion fidelity of the polymerase, the extension fidelity of the polymerase, and the removal efficiency of the exonuclease. We are currently doing experiments to address to what extent the different processes contribute to the fidelity, but these experiments lay beyond the scope of the current work.

Is there precedence for EndoIV to work on mismatches?

It was reported that Endonuclease IV from *Chlamydomonas reinhardtii* has 3'-5' exonuclease activity on DNA mismatches and that it may be required for proofreading function of Pol I during Okazaki maturation (Xie 2013). This however does not appear to be the case for *E. coli* EndoIV (Kerins 2003).

Xie JJ, Liu XP, Han Z, Yuan H, Wang Y, Hou JL, Liu JH. Chlamydomonas reinhardtii endonuclease IV prefers to remove mismatched 3' ribonucleotides: implication in proofreading mismatched 3'-terminal nucleotides in short-patch repair synthesis. DNA Repair. 2013 Feb 1;12(2):140-7

Kerins SM, Collins R, McCarthy TV. Characterization of an endonuclease IV 3'-5' exonuclease activity. J Biol Chem. 2003 Jan 31;278(5):3048-54

8. Electron density maps are provided in some figures. **These maps should be simulated annealing omit maps (and described as such) and the type of map (e.g. 2mfofc), sigma level, and carving should be noted.**

The maps have been replaced with simulated annealing omit maps and the map information is given in the figure legends and Material and Methods.

9. The number of ligands in the crystal structure is surprising. **Assuming three zincs, what are the other ones? Perhaps the B-factors for the zincs, as they are the most central, should be provided in their own right. What's the evidence for the other ligands, as the resolution is only 2.8 ?**

The number of ligands (63) listed in the table is for the number of *atoms*. These come from 3 zincs, 9 sulfate ions and 1 HEPES. At 2.8 Å the sulfate ions are easy discernable as strong positive density that remains positive after placing a water molecule followed by several rounds of refinement and rebuilding. The HEPES molecule was identifiable based on the shape, hydrogen binding partners, and the high concentration of HEPES (100 mM) in the crystallization buffer.

We now indicate the origin of the ligand atoms in Table 1 and included a separate entry for the B-factors of the Zn ions.

Reviewer #2 (Remarks to the Author):

This study by Banos-Mateos et al. describes the molecular details of a crystal structure for the majority of the DnaE1 high-fidelity replicase from *M. tuberculosis*, detailing the physical relationship between the polymerase and exonuclease domains. This story follows on from their previous study (Rock et al., 2015) in which they reported that the PHP domain acts as the proofreading exonuclease subunit for this polymerase, enabling it to perform high-fidelity synthesis.

The structural studies reveal that this replicase has a similar architecture to that of the previously reported and closely-related replication polymerases from other bacteria, such as PolC and PolII. Using sequence analysis and HDX, they predict a putative ss DNA binding channel that runs between the active sites of the polymerase and PHP subunits and propose its possible role in directing DNA during synthesis for the removal of misincorporated bases by the PHP subunit. They subsequently describe the zinc binding properties of this domain, including the preferential binding of zinc ions to the exonuclease and the inhibition of its activity by an excess of magnesium ions. They then show that there is a significant similarity between the active sites of this PHP domain and EndoIV. Finally, based on these collective insights, they propose a zinc-dependent mechanism for this PHP-exonuclease.

Major points

1. In figure 2, the structure of the PHP domain active site, including bound metal ions, is described. The structure of other PHPs are known, such as those from bacterial Pol Xs, and therefore **the authors should compare these different structures to the Mt PHP domain, even if it is a supplementary figure, and also reference these key papers as they were among the first to describe this fold having exonuclease activity.**

We have included these references in the text (refs 16 and 17). A comparison of *Thermus aquaticus* Pol III, *Deinococcus radiodurans* PolX and the inactive PHP domain from *E. coli* Pol III have been included as Supplementary Fig. 2.

2. Although they use structure, sequence and HDX analysis to identify a putative ss DNA binding channel. The data presented in figure 3 is anecdotal and circumstantial and only weakly supports their model. In particular, the sequence data in Figure 3a is very confusing and could mean a range of things so it is not particularly persuasive. **The authors need to provide much more functional biochemical / biophysical evidence to validate that this channel actually binds such substrates. For example, mutate the residues that line the channel and study the consequences on DNA binding and catalysis. Use crosslinking to show DNA binding in this region. Use FRET probes on DNA to demonstrates binding in the channel.**

As stated in the manuscript, we identify the narrow groove as a *potential* ssDNA binding site. We base this on the fact that the narrow channel forms the shortest, direct connection between the polymerase and exonuclease active site. The sequence conservation data shows that the narrow groove is conserved in C-family polymerases with an active PHP-exonuclease, but not in polymerases with an inactive PHP-exonuclease, which further supports an important role for this groove. The HDX data in addition reveals that the groove and area around are affected by the presence of DNA. Furthermore, a narrow ssDNA channel is also observed in several other nucleases as now noted in the main text (page 6, lines 12-13) and shown Supplementary Fig. 4. The ssDNA binding appears to be mimicked by the two SO_4^{2-} ions, in a manner similar to the two SO_4^{2-} ions in the polymerase active site (Fig 1d). Together, support a potential role in binding of the ssDNA substrate during removal of the misincorporated nucleotide.

In response to the reviewers suggestion, we have made multiple mutations in and around the narrow groove and PHP active site. Unfortunately, all mutations of conserved residues in the narrow groove resulted in insoluble protein or no protein expression (see reviewer2-figure 1 below). We therefore were not able to biochemically validate the role of the narrow groove. The suggested methods of crosslinking and FRET are both low resolution methods that are not able to discriminate between DNA binding within or close to the narrow groove. For this reason we opted not use these methods.

Reviewer 2-Figure 1. Mutation of conserved residues in the narrow groove result in insoluble protein or loss of protein expression. Non-permissive mutations are indicated in red, while permissive mutations are shown in cyan.

Ultimately, these models can only be properly validated when they have a structure with ss DNA bound, this key piece of the puzzle is currently lacking from this study.

We agree with the reviewer that the most detailed information about the potential ssDNA binding groove would come from a co-crystal structure. We have performed many crystallization trials with different DNA substrates (ssDNA, dsDNA with mismatch) and different versions of the protein (WT and mutants), but have not been able to obtain any co-crystals. This remains an important aim for us, for which we are currently expressing and purifying DnaE1 polymerases from other mycobacterial species in the hope that these might be more successful in yielding crystals with DNA bound. But this is a significant body of work that is beyond the remit of this work.

3. While it is clear from their studies that zinc ions preferentially bind in the PHP active site, they have not determined the K_D of binding for this or various other competing metal ions, which could be achieved by doing competition assays using a range of methods. This is essential because in figure 4e & f they show that magnesium ions inhibit the exonuclease activity.

As shown in Fig 4, the PHP-exonuclease retains three zinc ions throughout the protein purification and is fully active without addition of other metals. Therefore, to determine the affinity for Zn, we first removed the metals with EDTA, which results in a loss of activity (see Reviewer 2 - Fig 2, lane 3). This is irreversible, as addition of increasing amount of zinc does not restore activity. Therefore, we were not able to determine the K_D for zinc.

Next, we performed competition assays by addition of different metals (Zn, Mn, Ni, Co) to the PHP-exonuclease (see Supplementary Fig. 6). Here we find that exonuclease inhibition by the different metals is reached at 5-10 mM, similar to that of Mg. However, only Mg is found in mM concentrations in the cell, while the intracellular concentration for the other metals are far lower: 10^{-6} M for Mn, and 10^{-10} M for Zn, Ni, and Co (Foster *et al*, 2014). Therefore, inhibition of the PHP-exonuclease by the other metals is unlikely to occur *in vivo*.

Foster AW, Osman D, Robinson NJ. Metal preferences and metallation. *Journal of Biological Chemistry*. 2014, vol. 289, pp 28095–103.

Reviewer2-Figure 2. The WT protein show robust exonuclease activity without the requirement of extra metals (Lane 2). Addition of EDTA results in loss of exonuclease activity (Lane 3). Next, the protein was purified by gel filtration to remove the EDTA and increasing amount of Zn added back to the protein. However, titration of increasing amount of Zn does not restore exonuclease activity.

Although this is interesting, the next step is to show how this occurs and why, they have not investigated this finding in any significant way.

It would indeed be of interest to know the mechanism by which Mg inhibits the PHP-exonuclease. However, to us the important point is that magnesium, a metal that is present at mM concentrations in the cell and used by all DNA polymerases, is inhibitory to the PHP-exonuclease at concentrations that are physiological (1-10mM), and therefore a potential problem that the Mycobacteria need to deal with. The lack of a mechanism for how Mg causes inhibition does not lessen the impact of this observation.

4. The demonstration that the active site of PHP shows a "mirroring" of the zinc ions binding sites of EndoIV is interesting and based on the DNA-bound structures of this nuclease they propose a catalytic mechanism for this PHP. Although it is legitimate to do this, such models then need to be validated using experimental biochemical studies, e.g. classical kinetic approaches combined with site-directed mutagenesis, to evaluate the proposed nucleolytic cycle. It is essential that such studies are undertaken to back up the model presented in figure 7.

We have mutated the two amino acids that are key to our model: glutamate 73 and aspartate 226. Mutation of glutamate 73 to glutamine (i.e. replacement of one oxygen for a nitrogen) results in insoluble protein. This is consistent with a predicted structural role in coordinating two zinc ions that are important for the integrity of the PHP domain. In contrast, a similar mutation of aspartate 226 to asparagine does not affect protein expression or metal binding (Fig 4b), yet results in a complete loss of exonuclease activity (Rock *et al* 2015, Nat Genet). This is consistent with the predicted role of aspartate 226, where one oxygen ligands a zinc ion, while the second oxygen activates a water molecule. These results are now included in the section "*The PHP-exonuclease shows a striking similarity to Endonuclease IV*" (page 11, lines 6-11) and Figure 4b.

Rock JM, Lang UF, Chase MR, Ford CB, Gerrick ER, Gawande R, et al. DNA replication fidelity in *Mycobacterium tuberculosis* is mediated by an ancestral prokaryotic proofreader. *Nat Genet.* 2015, 47 pp677–81.

Minor Points

5. In figure 1, the residue numbers label should be labelled in the electron density maps, although I'm not sure it is really needed and this could go into the supplementary section.

The figure has been amended accordingly.

6. In figure 2, they refer to either "Zinc" or "Metal" to describe the bound ions, Given they have not yet shown it is zinc, demonstrating this later in the paper, maybe they should use the term "Metal" ?

This has been corrected.

Reviewers' comments:

Reviewer #2 (Remarks to the Author):

The authors have made only minimal changes to the manuscript and have failed to obtain the key experimental data required (see specific comments below) to address the queries regarding metal and ss DNA binding. Without this additional supporting data, the potential novel aspects of this study are speculative, with minimal supporting data to validate their models. The authors make major conclusions about the mode of DNA binding and the catalytic mechanism, the two novel parts of this story, based on the structure and modelling data rather than complementing these structural studies with more detailed experimental approaches to validate their hypotheses and conclusions.

"The suggested methods of cross-linking and FRET are both low resolution methods that are not able to discriminate between DNA binding within or close to the narrow groove. For this reason we opted not use these methods."

Maybe so but the onus is fully on the authors, not the referees, to come up with an experimental approach that addresses this key point otherwise, in the absence of a co-crystal structure, it is pure speculation...

Specific comments:

1. This point is adequately addressed.
2. Their HDX data is still vague. Most of the exchange is witnessed outside of the channel. No doubt this structural feature is well used, but the evidence is scant in this setting.

To disregard other techniques when addressing ss DNA interactions is surprising given they have been asked to do so, surely some kind of DNA cross-linking capture with MS verification would give some possible evidence.

Although they have failed to obtain co-crystals, how about making mutations to increase ss DNA binding?

3. The resolution of a mismatch followed by subsequent extension is a two-step process, their assay only examines the later step. Therefore, they need to look specifically at the inhibition of exonuclease activity alone. Reviewer 2-Figure 2, this is a more appropriate assay but it only shows EDTA-treated enzyme and then increasing zinc. Should also be performed with increasing amounts of other metal ions too. The results observed in Supplementary Fig 6 are most likely due to the effects of M²⁺ ions on extension.

4. This point has only been half addressed... with D226N, activity could theoretically be restored if activated water is titrated into the reaction mix (-OH ions). Likewise, shouldn't the activity be competitively inhibited by phosphate ions? An acid/base titration at various ionic strengths in the presence of phosphate ions should test this mechanism....

Reviewer #2 (Remarks to the Author):

The authors have made only minimal changes to the manuscript and have failed to obtain the key experimental data required (see specific comments below) to address the queries regarding metal and ss DNA binding. Without this additional supporting data, the potential novel aspects of this study are speculative, with minimal supporting data to validate their models. The authors make major conclusions about the mode of DNA binding and the catalytic mechanism, the two novel parts of this story, based on the structure and modelling data rather than complementing these structural studies with more detailed experimental approaches to validate their hypotheses and conclusions.

“The suggested methods of cross-linking and FRET are both low resolution methods that are not able to discriminate between DNA binding within or close to the narrow groove. For this reason we opted not use these methods.” Maybe so but the onus is fully on the authors, not the referees, to come up with an experimental approach that addresses this key point otherwise, in the absence of a co-crystal structure, it is pure speculation...

We agree with the reviewer that definitive information regarding the position of the primer strand during the exonuclease mode can only be obtained experimentally. We have therefore spent a considerable amount of time and effort to determine the entry path of the primer strand into the PHP exonuclease. Finally, we have used cryo-EM to image the complex of Mtb DnaE1, β -clamp, and a DNA substrate containing a mismatch, analogous to our previous work on the *E. coli* complex (Fernandez-Leiro 2015, *Elife* (PMID: 26499492) & Fernandez-Leiro 2017, NSMB (PMID: 28067916). However, due to a strong preferential orientation of the complex in the sample grid, we were not able to determine a three-dimensional cryo-EM map of the Mtb complex. Modifications of conditions, DNA substrate, protein, or grid type did not improve the preferred orientation. Fortunately, one of the 2D class averages provided a very clear view of the polymerase, clamp, and DNA (see Fig. 3a and b). Using this as a guide, we could then model the position of the polymerase, clamp, and DNA. This revealed that the primer strand does not travel through the narrow groove as proposed in the earlier version of the manuscript, but enters the active site under a different angle. We have confirmed this by mutational analysis of several residues that line the new entry path. Indeed, while these mutants show full polymerase activity on a matched DNA substrate, they show strongly suppressed activity on a mismatched DNA substrate that requires exonuclease activity prior to polymerase activity. The new model also fits well with our HDX analysis, as in the new entry path, the primer strand is in direct contact with a long loop in the PHP domain that is most affected by the presence of DNA in the HDX analysis. Hence, we believe that the model based on our 2D cryo-EM images and mutational analysis accurately describes the position of the primer strand in the exonuclease mode.

Specific comments:

1. (other PHP domains) This point is adequately addressed.

2. (ssDNA binding groove) Their HDX data is still vague. Most of the exchange is witnessed outside of the channel. No doubt this structural feature is well used, but the evidence is scant in this setting.

To disregard other techniques when addressing ss DNA interactions is surprising given they have been asked to do so, surely some kind of DNA cross-linking capture with MS verification would give some possible evidence. Although they have failed to obtain co-crystals, how about making mutations to increase ssDNA binding?

Please see comments above.

3. (use of other metals) The resolution of a mismatch followed by subsequent extension is a two-step process, their assay only examines the later step. Therefore, they need to look specifically at the inhibition of exonuclease activity alone. Reviewer 2-Figure 2, this is a more appropriate assay but it only shows EDTA-treated enzyme and then increasing zinc. Should also be performed with increasing amounts of other metal ions too. The results observed in Supplementary Fig 6 are most likely due to the effects of M²⁺ ions on extension.

We have now tested metal inhibition of exonuclease activity using a ssDNA substrate (Supplementary Fig. 6). In addition, we have also measured inhibition of polymerase activity on three different dsDNA substrates. All four metals tested (Zn, Mn, Ni, Co) inhibit the exonuclease at 2-10 mM concentration. Three of these (Zn, Ni, Co) also inhibit the polymerase at comparable concentrations. The only exception to this is Mn, that stimulates the polymerase (but not exonuclease) and enables mismatch extension, as shown by a mismatched DNA substrate with a non-cleavable phosphothioate bond. (This error-inducing effect of manganese on the fidelity of DNA polymerases is well known). Importantly, all four metal exists in the cell at levels that are 1000 - 100,000 times lower than needed to obtain inhibition. Therefore, these four metals are unlikely to play any role in the modulation of exonuclease (or polymerase) activity of Mtb DnaE1. This is in contrast to Mg that does exist in mM concentrations in the cell and inhibits the DnaE1 exonuclease at 5 mM Mg or higher.

4. (catalytic mechanism) This point has only been half addressed.... with D226N, activity could theoretically be restored if activated water is titrated into the reaction mix (-OH ions). Likewise, shouldn't the activity be competitively inhibited by phosphate ions? An acid/base titration at various ionic strengths in the presence of phosphate ions should test this mechanism....

We have measured the activity of the exonuclease at increasing values of pH (Fig. reviewer 1). We find that at pH 8.5 or higher, the activity of the WT exonuclease is inhibited. As a result, we were not able to restore the activity of the D226N mutant with increasing the concentration of OH⁻ ions.

Figure reviewer 1. The PHP-exonuclease is inhibited at higher pH. Gel electrophoresis of a 5' labeled single stranded DNA substrate after exonuclease digestion by WT and mutant (D226N) DnaE1 at increasing pH.

Because we have not been able to experimentally validate the role of D226N during catalysis, we have toned down our discussion regarding a possible mechanism for the PHP exonuclease. We now state that based on our comparison to Endonuclease IV, "...it is likely that in the Mtb PHP-exonuclease, aspartate 226 takes up the role of activating a water molecule for the nucleophilic attack on the phosphodiester bond of the bound DNA."

Reviewers' comments:

Reviewer #2 (Remarks to the Author):

1. To address the major issue regarding the location of the DNA in the PHP domain, which they were unable to determine using crystallographic or other means, the authors have used cryo-EM to try and answer this and come up with a 2D model based on class averaging. At first glance this looks promising, in that it is possible to dock in the clamp and polymerase. There is also additional density into which they have docked a DNA model, although without a "control" structure we have to take their word for this, see below. From this, they propose a new DNA docking site in a groove on the PHP surface, and not the one originally proposed in the original draft that they now disregard. To validate this model, they mutated residues lining the proposed "new" binding groove and show that these mutations alter the ability of the polymerase to extend mismatches but not matched DNA substrates.

However, while this may represent the possible binding site for DNA on PHP, it is far from proven. Firstly, as it is a 2D and not 3D model, it can be interpreted in a number of ways and the authors have not also presented a "control" structure with a matched DNA, as shown in their original model (Fig.1d) to help them ensure their model is likely correct. Having this model would enable them to more definitively show, using some kind of "difference density" mapping, that the additional density can be attributed to the DNA, and not some rearrangement in the protein structures. In figure 3d, they present the docking of the 3' end of the primer strand as if it was a crystal structure but it is only a model, the primer could bind in many different orientations so this is misleading and again could lead to an incorrect interpretation of the binding of the density as the precise 3D orientation of this strand cannot be derived from a 2D model.

Although their mutagenesis data appears to confirm the role of the proposed DNA binding residues in the docking of the primer strand, based on polymerase assays, the authors need to confirm that the mutations also disrupt/reduce its 3' nuclease activity on mismatches. It is surprising that these assays have not been performed given it is a nuclease after all. Although potentially compelling, the polymerization defects observed could be ascribed to other structural changes too so this is an important point to be addressed.

2. The authors measured the levels of inhibition given by increasing amounts of other divalent metals. The results suggest that the effect on exonuclease activity is roughly the same, no matter what ion is used, which is largely ignored (Supplementary figure 5). Could this be due to the fact that the Zn ions are non-exchangeable and the subsequent inhibitory effects are due to changes in ionic strength? The authors should place much more emphasis on the (possible) non-exchangeable nature of the Zn ions in the PHP domain and not fully exclude the possibility that other, maybe more transient, metal ions could also play roles as they have not been able to experimentally disprove this fact, due to the inability to exclude the zinc ions. For example, cobalt and manganese less inhibitory at high concentrations than zinc

3. The authors have toned the mechanistic discussion regarding the role of D226 and nucleophilic activation of water as they have not presented any experimental evidence to support this model, except the structural comparison with Endo IV. However, it is disappointing that the authors didn't properly perform the acid / base activity titration as requested comparing the WT and D226N mutant. Surely, at low pH (their assays only went from pH 7.5 upwards but not down) D226 would start to protonate and be less able to activate the water for nucleophilic attack?

Point-by-point response to reviewer #1.

The authors have addressed most of the reviewer concerns. However, the zinc overlays appear confusing. In Figure 5A, the alignment of the active site shows a mirror symmetry in the Zn positions. However, in the movie, the zinc ions directly overlay. This is likely due to the positioning of the leaving group. This is indirectly written in the text, but a more direct sentence would be insightful for readers.

We no longer present the movie in the latest version of the manuscript, as our cryo-EM analysis combined with HDX and mutational analysis has revealed that the original model for the entry path of the primer strand into the PHP-exonuclease was incorrect. The new DNA model for the exonuclease state is presented in Figure 3 of the current manuscript.

What does the overlaid side chains in the active site look like when the zincs are overlaid? This may be done in Figure 6c and d, although the zinc atoms are still now overlaid. Figure 6 is also confusing. Were the active sites of EndoIV and PHP-exo overlaid? The different panels appear to be in slightly different orientations. Could the spheres of the zinc positions be shown in Figure 6A?

As we no longer present this model, we have deleted former Figure 6 from the manuscript. For the reviewers information, we show the requested comparison of the two active sites with the metals overlaid below. This figure is rather confusing (even after leaving out the two residues that are located under the zinc ions), which is why we feel it would not add to the clarity of the manuscript.

Figure reviewer 1. Overlay of Endonuclease IV and PHP-exonuclease active site after superimposition of the three zinc ions. Endonuclease IV residues are shown in green sticks. PHP-exonuclease residues in grey sticks. Lower case 'e' or 'd' at the end of the residue label indicates Endonuclease IV or DnaE1, respectively.

Point-by-point response to reviewer #2

1. To address the major issue regarding the location of the DNA in the PHP domain, which they were unable to determine using crystallographic or other means, the authors have used cryo-EM to try and answer this and come up with a 2D model based on class averaging. At first glance this looks promising, in that it is possible to dock in the clamp and polymerase. There is also additional density into which they have docked a DNA model, although without a “control” structure we have to take their word for this, see below. From this, they propose a new DNA docking site in a groove on the PHP surface, and not the one originally proposed in the originals draft that they now disregard. To validate this model, they mutated residues lining the proposed “new” binding groove and show that these mutations alter the ability of the polymerase to extend mismatches but not matched DNA substrates.

However, while this may represent the possible binding site for DNA on PHP, it is far from proven. Firstly, as it is a 2D and not 3D model, it can be interpreted in a number of ways and the authors have not also presented a “control” structure with a matched DNA, as shown in their original model (Fig.1d) to help them ensure their model is likely correct. Having this model would enable them to more definitively show, using some kind of “difference density” mapping, that the additional density can be attributed to the DNA, and not some rearrangement in the protein structures.

We now present two alternative models in supplementary Fig 4: one with the DNA modeled in the 'polymerase' mode (similar to the model in Fig 1d and Supplementary Fig. 1), and one with the primer strand modeled to bind in the narrow groove that runs from the polymerase active site to the PHP-exonuclease active site. In both models, the DNA does not fit the extra density between the polymerase and clamp. For ease, we also show this figure below.

Supplementary Figure 4 | Modeling of the DNA into the 2D cryo-EM class. Different DNA models were fitted into the 2D cryo-EM class. (a) Model with the primer strand entering the PHP-exonuclease between short β -hairpin and long loop in the PHP domain (see Fig. 3). (b) DNA modeled in the polymerase mode (see Supplementary Fig. 1) (c) Model with the primer strand entering the PHP-exonuclease via the narrow groove between polymerase active site and exonuclease active site (see Fig. 2d).

In figure 3d, they present the docking of the 3' end of the primer strand as if it was a crystal structure but it is only a model, the primer could bind in many different orientations so this is misleading and again could lead to an incorrect interpretation of the binding of the density as the precise 3D orientation of this strand cannot be derived from a 2D model.

In this figure the path of the modeled DNA is indicated by a single blue line. To emphasize it is only a model we have omitted the bases and backbone atoms. For additional clarity, we have now included an extra line in the figure legend: "the path of the modeled DNA is indicated by the blue line".

Although their mutagenesis data appears to confirm the role of the proposed DNA binding residues in the docking of the primer strand, based on polymerase assays, the authors need to confirm that the mutations also disrupt/reduce its 3' nuclease activity on mismatches. It is surprising that these assays have not been performed given it is a nuclease after all. Although potentially compelling, the polymerization defects observed could be ascribed to other structural changes too so this is an important point to be addressed.

The reviewer suggests that our real time polymerase/exonuclease assays could be hiding 'other structural changes'. This is incorrect. The experiments we present measure BOTH polymerase and exonuclease activity by using either a matched or mismatched DNA substrate. Previously we have shown that in the absence of exonuclease activity, the polymerase cannot extend from a mismatched template (Rock 2015, Nat. Genetics vol 47, pp677-681). Our results show that for the mutants in the DNA path, ONLY the activity on a mismatched substrate is reduced, while the polymerase activity on a matched substrate is unaffected. This therefore negates the possibility that 'other structural changes' occur.

2. The authors measured the levels of inhibition given by increasing amounts of other divalent metals. The results suggest that the effect on exonuclease activity is roughly the same, no matter what ion is used, which is largely ignored (Supplementary figure 5). Could this be due to the fact that the Zn ions are non-exchangeable and the subsequent inhibitory effects are due to changes in ionic strength? The authors should place much more emphasis on the (possible) non-exchangeable nature of the Zn ions in the PHP domain and not fully exclude the possibility that other, maybe more transient, metal ions could also play roles as they have not been able to experimentally disprove this fact, due to the inability to exclude the zinc ions. For example, cobalt and manganese are less inhibitory at high concentrations than zinc.

We show in Fig. 4 and Supplementary Fig. 6 that different metals inhibit the PHP exonuclease at mM concentrations. Yet only Mg is found at mM concentrations in the cell, and is therefore the only metal that could affect the activity in vivo. This is relevant as Mtb encounters different magnesium concentrations in the hosts which could therefore affect the fidelity of DNA replication. The reviewer would like to see a more in-depth analysis/discussion of how the inhibition by the different metals is achieved. Such an analysis would not strengthen our

observation and would be of little interest the broader audience of *M. tuberculosis* and/or DNA polymerase researchers. We therefore feel that such an in depth analysis into the mechanism of metal inhibition is outside the scope of our manuscript.

3. The authors have toned the mechanistic discussion regarding the role of D226 and nucleophilic activation of water as they have not presented any experimental evidenced to support this model, except the structural comparison with Endo IV. However, it is disappointing that the authors didn't properly perform the acid / base activity titration as requested comparing the WT and D226N mutant. Surely, at low pH (their assays only went from pH 7.5 upwards but not down) D226 would start to protonate and be less able to activate the water for nucleophilic attack?

In our manuscript, we *propose* that D226 may be the catalytic residue that activates a water molecule to perform the nucleophylic attack on the phosphate backbone. Previously, the reviewer asked to perform an exonuclease assay at increasing pH, as it could possibly to rescue the exonuclease activity of the D226N mutant with free hydroxyls present at higher pH. We saw the value of such experiments, as it would strengthen our assumption that the exonuclease reaction works via an hydroxyl. Yet, we found that at higher pH values the wild type protein looses activity, which therefore made it impossible to rescue the mutant (which showed no activity over the entire range tested). Now the reviewer asks us to repeat the experiments at lower pH values, in order to inhibit the exonuclease. Such an experiment would provide no insight into the mechanism of action as there could be numerous reasons why at lower pH the exonuclease would be inhibited. The three zinc ions in the PHP exonuclease are coordinated by nine residues, eight of which will be affected by a lower pH. Hence, lowering the pH to inactivate the PHP exonuclease would be an experiment of little use.